# The Influence of Graphene Oxide on Nanoparticle Emissions during Drilling of Graphene/Epoxy Carbon-Fiber Reinforced Engineered Nanomaterials

**Kristof Starost and James Njuguna *** 

Centre for Advanced Engineering Materials, School of Engineering, Robert Gordon University, Aberdeen AB10 7GJ, UK; k.starost@rgu.ac.uk
* Correspondence: j.njuguna@rgu.ac.uk

**Abstract:** Graphene oxide (GO) nanoparticles are increasingly being used to tailor industrial composites. However, despite the advantages, GO has shown conceivable health risks and toxicity to humans and the environment if released. This study investigates the influence that GO concentrations have on nanoparticle emissions from epoxy-reinforced carbon fiber hybrid composites (EP/CF) during a lifecycle scenario, that is, a drilling process. The mechanical properties are investigated and an automated drilling methodology in which the background noise is eliminated is used for the nanoparticle emissions measurements. Real-time measurements are collected using a condensation particle counter (CPC), a scanning mobility particle sizer spectrometer (SMPS), a real-time fast mobility particle spectrometer (DMS50) and post-test analytical methods. The results observe that all three nanoparticle reinforced samples demonstrated a statistically significant difference of up to a 243% increase in mean peak particle number concentration in comparison to the EP/CF sample. The results offer a novel set of data comparing the nanoparticle release of GO with varying filler weight concentration and correlating it the mechanical influence of the fillers. The results show that the release characteristics and the influence in particle number concentration are primarily dependent on the matrix brittleness and not necessarily the filler weight concentration within the nanocomposite.

**Keywords:** nanoparticle; release; graphene; oxide; drilling; epoxy; nanocomposite; carbon-fiber

## 1. Introduction

Over the last couple of decades, epoxy-based (EP) nanocomposites have undergone intensive research and development ensued by their increasing implementation within commercial applications [1]. Carbon-fiber (CF) reinforced-EP composites have become well-established materials within various lightweight applications, most prominently the aeronautical and automotive industry. Nano-sized graphene oxide (GO) has been shown to improve and tailor the EP/CF matrix mechanical properties, as demonstrated in numerous studies [2–5]. However, the unique advantages and influence that GO nanoparticles have on the material properties accompany potential exposure to unique toxic effects within biological systems. Carbon-based nanofillers are of particular interest within the nanotoxicity literature due to the exceptional material properties and wide use across industries. Consequently, GO has demonstrated potential toxicological and/or eco-toxicological hazard [6–11]. Although the nanoparticles are initially embedded within the nanocomposites, a full understanding of the inadvertent release of nanoparticles within the workplace and atmosphere poses unknown potential hazards which are yet to be fully understood or quantified [12].

EP is one of the most utilized thermosets within polymers, with an estimated global EP resin market to be USD 10.6 billion by 2023 at a compound annual growth rate of 5.24% between 2017 and

2023 [13]. A separate report on global fiber reinforced composites forecasts a compound annual growth rate of 8.20% between 2018 and 2024 [14]. The use of carbon nanofillers to improve interfacial bonding between CFs and the polymer matrix is widely demonstrated in the literature with the use of GO [3,15,16].

Throughout its use, a polymer nanocomposite will undergo industrial machining where drilling, along with other machining scenarios, can lead to material damage and/or the release of potentially toxic nanoparticles [17,18]. Three excellent review studies on the release and/or exposure of nanoparticles due to lifecycle scenarios on nanocomposite materials have highlighted similar findings that high-quality evidence has demonstrated nanoparticle exposure, which is relevant during machining [18–20]. One study [19] concluded that whilst data currently indicate a high portion of the release to be partially or fully embedded nanomaterials, there is a shortage of research into the release of manufactured nanomaterials. In another study [18], the authors concluded that although there is a lack of data for engineered nanomaterial (ENM) exposure and with limitations between data, the literature suggests that all three routes of exposure (i.e., inhalation, dermal and ingestion) are relevant for workers in the manufacturing of ENMs.

One potential route for exposure is during the composite and component manufacturing stages involving processes such as drilling for joining, integration and assembly of parts [21]. For example, approximately 180,000 holes are drilled to produce a single Airbus A380 wing, and around 60% of rejected parts are due to defects introduced in holes [22]. Composites drilling is therefore an important operation at the manufacturing stage that leads to the generation of dust and potential nanoparticle release. From the current literature, only nine studies [23–31] have been identified to have investigated the nanoparticle release from drilling in nanocomposite materials. All nine of the studies on drilling demonstrated nanoparticles to be released. However, whilst some studies demonstrated a 56-fold increase in nanoparticle release due to the addition of $SiO_2$ [26], or a 211% increase with the addition of CNTs [29], some studies showed smaller influences, such as one study [27] which utilized a 0.09 wt.% $SiO_2$ in polyurethane and only observed a minimal increase in particle number concentration. Noticeably though, studies have not investigated the influence of varying weight concentration of the nanoparticle filler on nanoparticle release. Nonetheless, most studies concluded the nanoparticle fillers to have a statistically significant influence on the nanoparticle release, but observed significant differences between studies in the particle number concentrations. The findings from the nine studies on the influence of nano-sized fillers on nanoparticle release during drilling are in general agreement with the lack in knowledge and the need for a harmonized methodology in order to compare the materials and processes that are reported in the literature on nanoparticle release from machining.

The aim of this study is therefore to investigate the influence GO has on nanoparticle release from EP/CF hybrid composites during drilling. As EP/CF composite materials are currently mostly used within the automotive and aeronautical industry [14], the materials will undergo drilling during the assembly and manufacturing stages. As evident within several studies, composite materials with nanoparticles have shown potential release of the nanoparticles [18–20]. This study will therefore investigate the influence of the GO nanoparticles on the nanoparticle release during the identified release scenario, drilling.

## 2. Experiments

### 2.1. Materials

Epoxy composites are vastly reported and used within industry to be reinforced with more conventional, micron-sized carbon fiber. The use of GO and other carbon-based nanofillers incorporated into hybrid CF and EP composites is less well known and embeds various nanofiller concentrations [32–34]. For this reason and based on studies within the literature [3,35–37], GO was chosen as a filler with concentrations of 0.05, 0.1 and 0.5 wt.%.

A commercially available high-performance bisphenol-A-(epichlorhydrin)-based epoxy resin specifically formulated for use in vacuum resin infusion from Easycomposites, Stoke on Trent,

UK (IN2 Epoxy Infusion Resin) combined with a polyoxypropylendiamin-based hardener from, Easycomposites, Stoke on Trent, UK (AT30 Epoxy Hardener –Slow) was chosen for the matrix. Graphene oxide (GO) flakes, 15–20 sheets with 4–10% edge-oxidized from Sigma-Aldrich, Dorset, UK (796034 Aldrich) were employed in this investigation. The 3k 2/2 twill woven carbon fiber was obtained from Easycomposites, Stoke on Trent, UK (Carbon Fibre 2/2 Twill 3k 210g).

The composite samples were manufactured through the vacuum resin infusion method [5]. Concentrations of 0.05, 0.1 and 0.5 wt.% relative to the wt.% epoxy were initially dispersed within methanol with the use of a sonication bath for 1 hour to allow for later dispersion of the GO in the epoxy. Once fully dispersed, the solution was then homogenously dispersed within the bisphenol-A-(epichlorhydrin)-based epoxy and placed in a vacuum oven for 2 hours at 60 °C to allow for slow solvent evaporation. The solution was then mixed together with the hardener using a magnetic stirrer and manual mixing. This was followed by the vacuum resin infusion process with 6 layers of the carbon fiber textile layered within a mold and left to cure for 7 days at room temperature. A 60:40 fiber-epoxy volume ratio and epoxy resin to hardener (100:30) was employed as recommended by the supplier. A reference sample without any GO was also manufactured (EP/CF), along with 0.05 wt.% GO (EP/CF/GO 0.05 wt.%), with 0.1 wt.% GO (EP/CF/GO 0.1 wt.%) and with 0.5 wt.% GO (EP/CF/GO 0.5 wt.%) samples.

*2.2. Automated Drilling Nanoparticle Release Methodology*

The methodology utilizes a process-related approach. This process is designed to simulate mechanical drilling on nanocomposite materials and is continued work from the Nanomaterials-Related Environmental Pollution and Health Hazards Throughout Their Life-Cycle (NEPHH) project study [25,26] funded by The European Commission (EC FP7 Project- CP-FP; Project Reference: 228536–2) and documented in three other studies [28–30]. A crucial factor identified in the literature and the Organisation for Economic Co-operation and Development (OECD) guidelines is for the methodology to make a distinction from the background or to control the background particles to setup a controlled environment. Building on the NEPHH project, the chamber is capable of achieving a clean environment monitored using a condensation particle counter (CPC), importantly removing all background noise or interference in the measurement of number concentration and particle size distribution. The data collected are therefore a representation of the particles released solely from the material. Removing the background data allows for a depiction of any particles released from the materials which can be directly linked as an unconditional maximum exposure assessment [38]. As proposed in several studies [39–41] with a controlled testing setup and environment, only one parameter, the material, is changed and investigated. This simplifies the issue of accounting for local background influences, as specified within the guidelines and reports by [42,43].

Once the chamber was cleared of any particles, the drilling studies were carried out by drilling across the width of the sample resulting in eight separate holes and bearing a time duration of 3 min of drilling, followed by 1 min of post-drilling. The eight holes drilled per sample were repeated three times to get an average of the particle number concentration and particle size distribution released.

Based on previous studies carried out on nanocomposite drilling [24–26], a standard Dremel 4000 drilling tool with an industrial standard stainless steel 3.5 mm twist drill bit was used at 10,000 rpm with a feed rate of 78 mm/min. The setup uses an automated drilling assembly operated externally to the chamber to permit a repeatable and controlled environment within the chamber as shown on Figure 1.

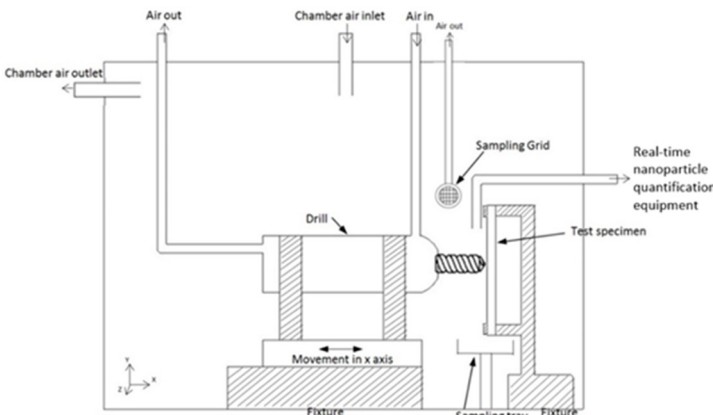

**Figure 1.** Design drawing of drilling setup within enclosed test chamber with cycled airflow to allow for a clean environment removing any background interference represented (not to scale).

The closed steel chamber has dimensions of 740 mm × 550 mm × 590 mm, and therefore a total inner volume of 0.240 $m^3$. It is designed to assure a closed environment to simulate an appropriate volume around the drill and minimizing electrostatic attraction to the surfaces. To quantify only the particles released from the sample, the chamber was initially cleared of particles through an inflow of clean air with the use of TSI 99.97% retention HEPA Capsule Filters. A separate capsule was constructed around the drill with separate air flow to avoid any interference of the drilling fumes on the particle number concentration within the capsule. The clean air system using the HEPA capsule filters and with the drill on was capable of producing a particle number concentration reading within the chamber of 0 particles/$cm^3$ with false background counts < 0.01 particles/$cm^3$, as measured using a TSI Environmental Condensation Particle Counter (CPC) model 3783 at a flow rate of 0.6 L/min, particle range of 7–3000 nm and concentration range of 0–$10^6$ particles/$cm^3$ and ± 10% at $10^6$ particles/$cm^3$. The level of background noise is therefore significantly within the ISO [44] cleanroom standard for particles ≥ 0.1 μm of 10 particles/$cm^3$.

An outlet channel is placed adjacent to the test specimen for the nanoparticle release equipment readings. A standard IOM Inhalable Sampler for the collection of inhalable particles was placed next to the test specimen with a 2 L/min suction to attract and prevent particles from detaching away from the grid for post-test chemical analysis [45]. An additional sampling tray was positioned below the test specimen for collection of the deposited particles for further post-test analysis.

The scanning mobility particle sizer (SMPS) used for the study is a TSI 3080 Electrostatic Classifier utilizing a nano differential mobility analyzer (DMA) with 99 distinct particle diameters within a particle range of 4.61–156.8 nm and a flow rate of 0.31 L/min. The principle of the Model 3080 Electrostatic Classifier with the DMA is based on the monotonic relationship between electrical mobility and particle size with singly charged particles. The aerosol particles go through a process of bipolar charging or "neutralization" and are then classified with the differential mobility analyzer and then measured by a condensation particle counter. The given particle size distribution is therefore corresponding to the electrical mobility diameter. In addition, separate repeated runs were carried out using a Cambustion DMS50 Fast Particle Size Spectrometer with a 1 s sampling period, inlet flow rate of 6 L/min, with 34 distinct particle diameters of size range between 4.87 and 562.34 nm for the particle size distribution. The DMS50 utilizes a unipolar corona charger placing positive charges on each particle which are then classified along electrometer detectors based on mobility and hence particle size [46]. The charge is conducted via an electrometer amplifier whose output indicates the flux of particles giving the particle concentration at that given particle size. Since the classification of particles according to their differing electrical mobility takes place in parallel (rather than in series as in the SMPS), the DMS50 can offer the faster sampled particle size distribution. This allowed for a size distribution every second compared to the SMPS of 45 s period (followed by 10 s for the classifier

to regenerate to its initial voltage and 5 s to start the size distribution again) and therefore a faster representation of the particles being released from the sample during drilling. The SMPS uses the assumption of spherical particles. Hence, from the diameters of the particle size distribution measured, and the material density of the nanocomposites, the particle mass size distribution can be estimated.

Both the Zeiss EVO LS10 Variable Pressure Scanning Electron Microscope and an SEM/EDX (FEI Quanta 200F) with a beam current of 208 μA and voltage of 10 kV were used for the present study and cross-checked using an electron probe microanalyser (EPMA) JEOL JXA-8621MX, with a beam current of 30 nA and voltage of 15 kV. SEM samples of the materials were prepared using sputter coating of an ultra-thin coating of gold to minimize charging. A sampling tray placed immediately below the drilling set up in the chamber was used to collect debris removed from the nanocomposites during the drilling operation.

### 2.3. Mechanical Properties

The influence of the addition of the GO nanofillers on mechanical properties (tensile and flexural) was evaluated and compared. To achieve this, the materials underwent a tensile test in accordance to ASTM D 3039/D tensile test [47] and 3-point flexural test in accordance with reference standard ASTM D 7264/M flexural test [48]. The tests were carried out with the use of an Instron 3382 universal testing system with a 100 kN load range. Raw data were collected using the Bluehill 3 software as measured in terms of load and extension. As per the respective standards, a constant head-speed of 2 mm/min for the tensile test and 1 mm/min for the flexural test was used, and data were collected at 10 Hertz. The data were then converted from the load and extension to stress vs. strain.

## 3. Results

### 3.1. Filler Effect on Mechanical Properties

Following the tensile and flexural testing standards of polymer matrix composite materials, the respective tensile and flexural properties of the materials were determined. A comparison of the tensile and flexural properties is shown in Figure 2.

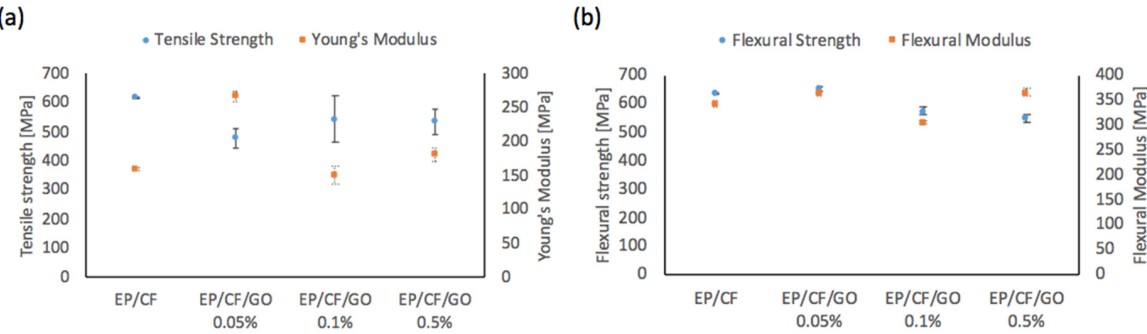

**Figure 2.** Comparison of mean values of epoxy-reinforced carbon fiber (EP/CF)-based samples reinforced with graphene oxide (GO) of (**a**) tensile properties and (**b**) flexural properties.

Statistical analysis was carried out on the samples. From a t-test between each sample and the EP/CF sample, only the EP/CF/GO 0.1 wt.% did not return a statistically significant difference for Young's modulus (EP/CF/GO 0.05 wt.% $p = 0.0000567$, EP/CF/GO 0.1 wt.% $p = 0.240$, and EP/CF/GO 0.5 wt.% 2 wt.% $p = 0.0391$). The analysis on the tensile strength, returned all of the reinforced samples with a statistically insignificant difference (EP/CF/GO 0.05 wt.% $p = 0.0814$, EP/CF/GO 0.1 wt.% $p = 0.222$, and EP/CF/GO 0.5 wt.% 2 wt.% $p = 0.175$). This is attributed to the high deviation between samples. Therefore, whilst none of the samples returned a statistically significant difference in tensile strength, only the EP/CF/GO 0.1 wt.% did not return a statistically significant difference for Young's modulus.

As with the tensile results, statistical analysis was carried out on the flexural tests. From the t-test between each sample and the EP/CF, all of the samples returned a statistically significant difference in flexural modulus (EP/CF/GO 0.05 wt.% $p = 0.00784$, EP/CF/GO 0.1 wt.% $p = 0.000622$, and EP/CF/GO 0.5 wt.% 2 wt.% $p = 0.0142$). In the analysis of the flexural strength, only the EP/CF/GO 0.05 wt.% returned with a statistically insignificant difference (EP/CF/GO 0.05 wt.% $p = 0.0575$, EP/CF/GO 0.1 wt.% $p = 0.0106$, and EP/CF/GO 0.5 wt.% 2 wt.% $p = 0.00765$). Therefore, whilst the EP/CF/GO 0.05 wt.% returned with a statistically insignificant difference in flexural strength, the inclusion of GO returned statistically significant differences in flexural modulus and flexural strength, both negative and positive.

The incorporation of the GO nanofiller had contrasting influence on the material properties. The influence of GO on mechanical properties will be correlated to the nanoparticle release.

### 3.2. Filler Effect on Particle Number Concentration

The GO-reinforced EP/CF samples underwent the repeated drilling, and the particle number concentration was measured during the testing. Using the CPC, the particle number concentration was quantified in situ with a sampling rate of 1 s. An average of the repeated test ($n = 3$) for each sample is displayed in Figure 3.

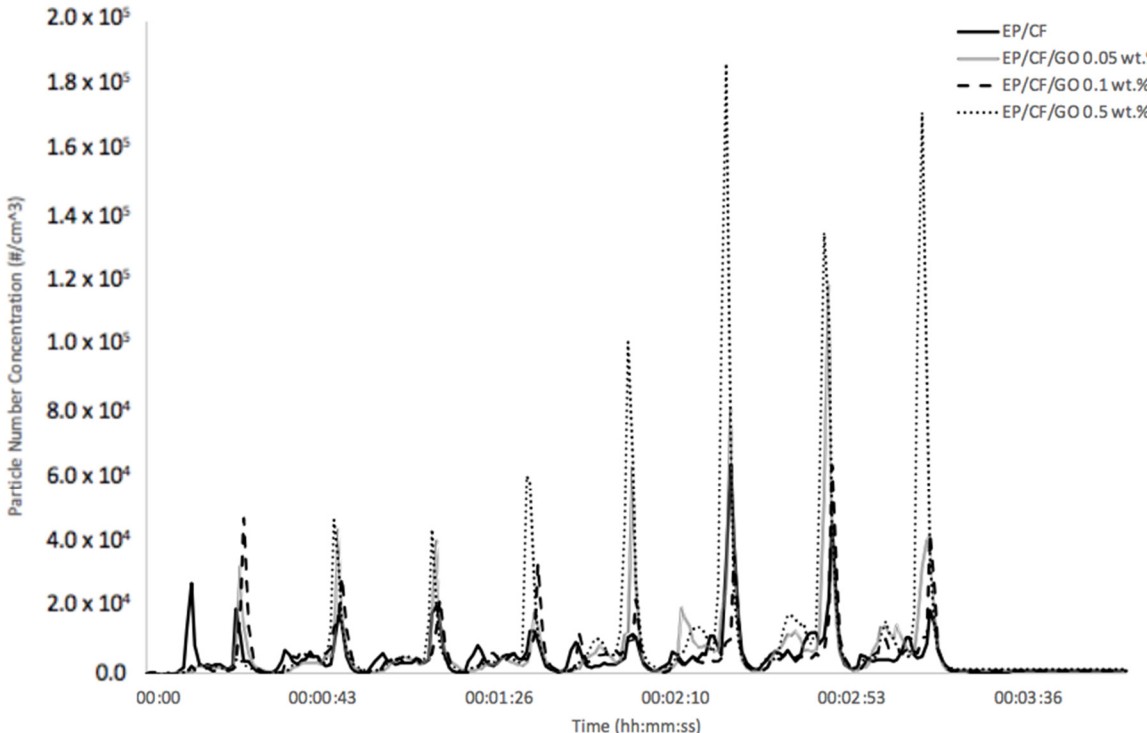

**Figure 3.** Particle number concentration averages of nanoparticles introduced from EP/CF-based samples reinforced with GO and measured using condensation particle counter (CPC) ($n = 3$ for each average).

The eight holes drilled are clearly evident within the particle number concentration over the 4 min of data sampling. Eight peaks represent the eight holes drilled, followed by one minute on no drilling and the concentration stabilizes. A maximum value in terms of quantity of the particles being released at the time of drilling is obtained. The methodology is able to provide a comprehensive depiction of the particles released during the drilling before (anything within 1 s) dispersion and scattering within the chamber. The particle number concentration can then be observed to relatively stabilize during the 1 min after the drilling has ended but does not drop back to the initial 0 #/cm$^3$. Thus, the particles produced from the drilling remain airborne within the chamber environment. These two meaningful annotations

therefore epitomize the release characteristics observed: peak particle number concentrations and remaining airborne concentration after 4 min of sampling.

The mechanical drilling therefore generates a substantial quantity of nanoparticles, which then quickly disperse, but remain airborne within the chamber (evident through stable concentration). From the comparison between the EP/CF sample and the reinforced samples with GO, any disparity between the samples is not clearly apparent.

The peaks concentrations of release during drilling are spread across two peaks which can be associated to the drill bit entering and the withdrawal of the drill bit from the material. The withdrawal of the drill bit can clearly be evident of producing the higher particle number concentration. Within the averages, only the first hole of the EP/CF samples displayed a higher particle number concentration from the drill bit entering the material than during the withdrawal. However, the introduction of GO into the samples at the three different weight concentrations did not demonstrate any noticeable difference to the profile of the release during entering or withdrawing the drill bit.

The data demonstrate that even the samples without the reinforcement of GO nanoparticles, released a substantial particle number concentrations during drilling. The EP/CF sample without any nanoparticles, observed a peak particle number concentration comparable to the samples reinforced with the GO nanoparticles. However, from the average profiles, the EP/CF/GO 0.5 wt.% sample can be seen to have obtained the highest peaks. These also can be seen to slowly increase in size over the eight holes, with the exception of the seventh hole. This would suggest the more holes being drilled also increases the particle number concentration peak size. Therefore, GO at 0.5 wt.% shows an increasing trend with more holes drilled, which is not evident within other studies [28–30].

Whilst the 0.5 wt.% GO is understood to increase the peak particle number concentrations released during drilling (with a 243% increase in mean peak particle number concentration), the two other GO weight concentrations have contrasting effects. The peaks introduced from the EP/CF/GO 0.1 wt.% followed a comparable profile to the peaks from the EP/CF samples, whereas the EP/CF/GO 0.05 wt.% released some slightly higher peak concentrations. A numerical and statistical representation of the data from all samples is shown in Table 1.

**Table 1.** Inferential statistical representation of the particle number concentrations introduced at the peaks due to the drilling on EP/CF-based samples reinforced with GO (*n* = 24 for each sample). Lower and upper limits represent the 90% confidence interval on a sampling t-distribution.

| Sample | Mean: $\bar{\bar{X}}$ (#/cm$^3$) | Deviation: $S_{\bar{X}}$ (#/cm$^3$) | Minimum (#/cm$^3$) | Maximum (#/cm$^3$) | 5% Lower Limit of Confidence Interval (#/cm$^3$) | 95% Upper Limit of Confidence Interval (#/cm$^3$) |
|---|---|---|---|---|---|---|
| EP/CF | $2.74 \times 10^4$ | $1.81 \times 10^4$ | $1.21 \times 10^4$ | $6.38 \times 10^4$ | $1.84 \times 10^4$ | $3.65 \times 10^4$ |
| EP/CF/GO 0.05 wt.% | $5.44 \times 10^4$ | $3.30 \times 10^4$ | $1.42 \times 10^4$ | $11.9 \times 10^4$ | $3.79 \times 10^4$ | $7.09 \times 10^4$ |
| EP/CF/GO 0.1 wt.% | $3.72 \times 10^4$ | $1.39 \times 10^4$ | $2.26 \times 10^4$ | $6.40 \times 10^4$ | $3.03 \times 10^4$ | $4.42 \times 10^4$ |
| EP/CF/GO 0.5 wt.% | $9.39 \times 10^4$ | $6.59 \times 10^4$ | $4.29 \times 10^3$ | $18.7 \times 10^4$ | $6.10 \times 10^4$ | $12.7 \times 10^4$ |

Table 1 displays the statistical analysis carried out on the peak particle number concentrations of the samples. The calculated lower tail of 5% and upper tail of 95% give a representation of the data for a 90% confidence interval of a t-distribution. This highlights the disparities between the peak particle number concentrations and therefore, a statistically significant difference with the introduction of GO on release in comparison to the EP/CF. A two-sample t-test of significance of each sample mean and deviation to the neat EP/CF sample returned statistically significant differences for all concentrations of GO (outside the 90% confidence interval). ANOVA single-factor analysis was performed to assess the variability between the sample peak means introduced due to the fillers. The analysis returned statistically significant differences within the four samples (F value = 4.63 F critical value = 2.95) and a 0.946% chance that the observation could have been observed due to random error alone, therefore rejecting a hypothesis that the samples displayed no difference.

It is important to note that although the statistical analysis returned a statistically significant difference with the introduction of the GO, this does not embrace the extent of the difference. From the data represented in both Table 1 and Figure 3, the incorporation shows a minor influence on the increase in particle number concentration. With the comparison of the samples, the EP/CF/GO 0.5 wt.% demonstrated a clear increase in all aspects of the particle number concentration, whereas the 0.05 and 0.1 wt.% displayed a more minor increase in peak particle number concentration values. As with all other samples, the statistical analysis does consider the high standard deviation and range demonstrated and therefore includes the level of randomness and variability in the peaks released.

From the numerical values, the EP/CF/GO 0.5 wt.% reinforced sample exhibited the uppermost mean value over the 4 min with $1.07 \times 10^4$ #/cm$^3$ introduced into the chamber during drilling. In relation to the EP/CF, the EP/CF/GO 0.05 wt.% and EP/CF/GO 0.1 wt.% produced a difference in nanoparticle concentration of 31.9% and $-1.17\%$ on average over the 4 min, respectively. Therefore, although the EP/CF/GO 0.05 wt.% and EP/CF/GO 0.5 wt.% observed an increase in particle number concentration over the 4 min, the EP/CF/GO 0.1 wt.% demonstrated a slight decrease.

Furthermore, to correlate the increasing weight concentration of GO on nanoparticle release, no statistical model can be created. This is due to an increase in concentration with 0.5 wt.% GO followed by a decrease from the 0.1 wt.% GO, and finally a larger increase from the 0.5 wt.% sample. The correlation therefore does not follow a trend or correlation between weight concentration and particle number concentration released.

It is notable that while the EP/CF/GO 0.5 wt.% sample produced the highest concentration and peaks during the drilling, it also demonstrated the highest concentration at the end of the four-minute sampling period. The high particle concentration introduced during the drilling indicates that the particles disperse within the chamber but crucially remain airborne. The EP/CF/GO 0.5 wt.% sample presented a particle number concentration remaining above $1.2 \times 10^3$ #/cm$^3$ even after the drilling and 1-min post drilling was concluded. The graphical representation of the average particle number concentration measured at the end of the four-minute sampling period is presented in Figure 4.

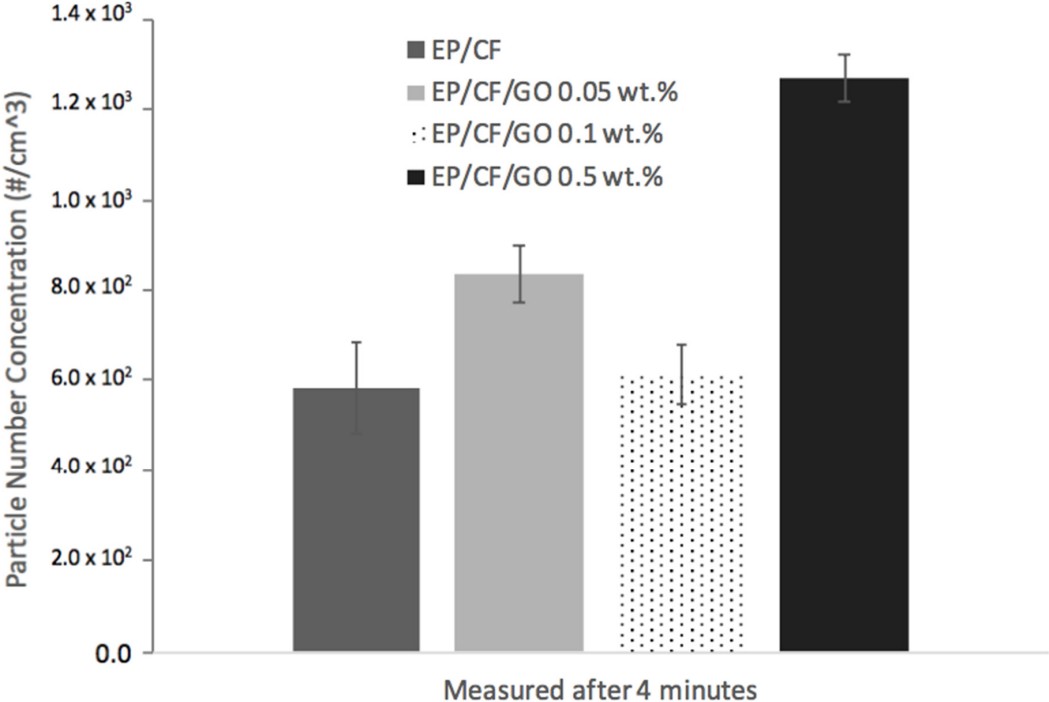

**Figure 4.** Particle number concentration recorded at 4th min (#/cm$^3$) for EP/CF-based samples reinforced with GO as measured on the CPC (*n* = 3 for each average).

The particle number concentration measured at the end of the sampling period is beneficial to evaluate the effect of the filler on the rapidity of depositing and dispersion within the chamber. The difference in particle concentration at the time of release due to the holes and concentration at the end of the sampling period presents an indication into these properties. The EP/CF/GO 0.5 wt.% sample observed a 118% increase measured at the 4th min from the EP/CF sample in comparison to the 112% increase over the previous four minutes (EP/CF/GO 0.05 wt.% increase of 44% and EP/CF/GO 0.1 wt.% increase of 5% from the EP/CF sample measured at the 4th minute). The difference at the 4th minute being similar to that measured over the four minutes demonstrates that the deposition rate during the 1-min post drilling is similar between all samples. Therefore, as well as demonstrating the highest peak particle number concertation released during drilling, the particles released from the EP/CF/GO 0.5 wt.% do not deposit any quicker and remain airborne to observe the highest particle number concentration post drilling.

### 3.3. Filler Effect on Particle Size Distribution

Simultaneously to the data gathered for the particle number concentration, the particle size distribution was quantified in situ using the SMPS and the DMS50. This provides a better understanding of the size of the particle number concentration seen in the Figure 3. An average of the four 1-min sampling periods measured across the four minutes is represented in Figure 5.

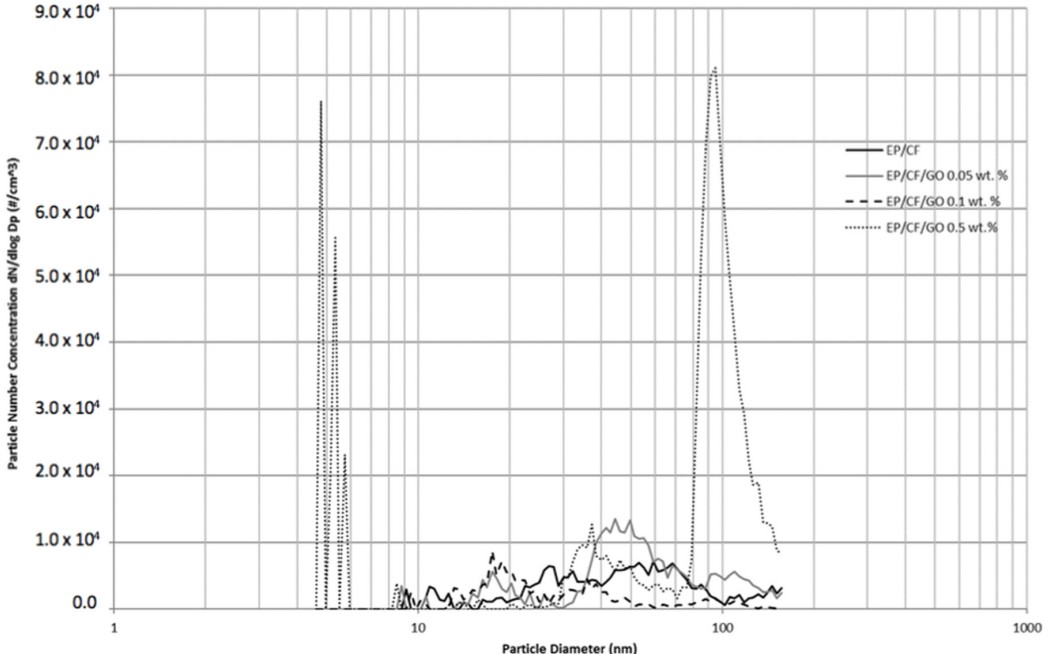

**Figure 5.** Average particle size distribution measured using scanning mobility particle sizer (SMPS) of EP/CF-based nanocomposites reinforced with GO (*n* = 12 for each average).

From the distribution, the EP/CF/GO 0.5 wt.% can be seen to have the most substantial effect on the particle size distribution. Two large peaks are observed on the limits of the SMPS between particle diameters of 4 to 6 nm and 80 to 100 nm. All the other samples observed smaller peaks, and at different particle diameters. The size distribution illustrates minimal effect with the introduction of GO nanofillers at 0.05 and 0.1 wt.% in comparison to the EP/CF sample. Excluding the EP/CF/GO 0.5 wt.%, the size distribution shows a relative scatter across the 100 nm spectrum. Slight increases are observed at 18 nm and between 40 and 50 nm, but these are still unparalleled compared to the peaks observed from the EP/CF/GO 0.5 wt.% sample.

The peaks observed below 6 nm from the EP/CF/GO 0.5 wt.% are quite significant in magnitude and substance. The GO embedded within the EP/CF consists of 15 to 20 sheet flakes which will therefore have a thickness of up to 20 nm. Each GO sheet can have a thickness of around 1 nm (796034 Sigma Aldrich, Dorset, UK). Drilling creates shear forces within the material which can therefore be related to possible separation of the layers due to the drilling. Furthermore, the EP/CF sample without any nanofiller did not exhibit any release peaks at these diameters. It is possible therefore, but only as a presumption, that the peaks observed below 6 nm could be associated to the GO fillers. However, this cannot be confirmed without identification of the independent GO fillers and peaks at the original thicknesses of the GO would be expected at around 20 nm.

The peak observed around 100 nm does not correlate with either of the individual fillers. The CF fibers have a thickness within the micron-range and were not apparent in the particle size distribution of the EP/CF sample. Any independent CF or matrix-filler (EP and CF) embedding released from the samples would be expected within the EP/CF sample. The peak may be associated to either agglomerations of the GO nanoparticles or GO embedded within the matrix. However, both would also be expected within the other GO reinforced samples, unless the higher weight concentration is likely to increase the separation of the GO from the CF. Nonetheless, the EP/CF/GO 0.5 wt.% is deduced to have influenced the particle size distribution quite significantly. In comparison, however, the EP/CF/GO 0.05 wt.% and EP/CF/Go 0.1 wt.% observed minimal influence on the particle size distribution in contrast to the EP/CF sample.

Further to the data collected on the SMPS, separate data were gathered on the DMS50 for the size distribution at each second and is displayed in a 3D plot as demonstrated for the 0.1 WT.% sample in Figure 6 (Note: data were taken from a separate run to the CPC and SMPS data due to the required increased inflow rate).

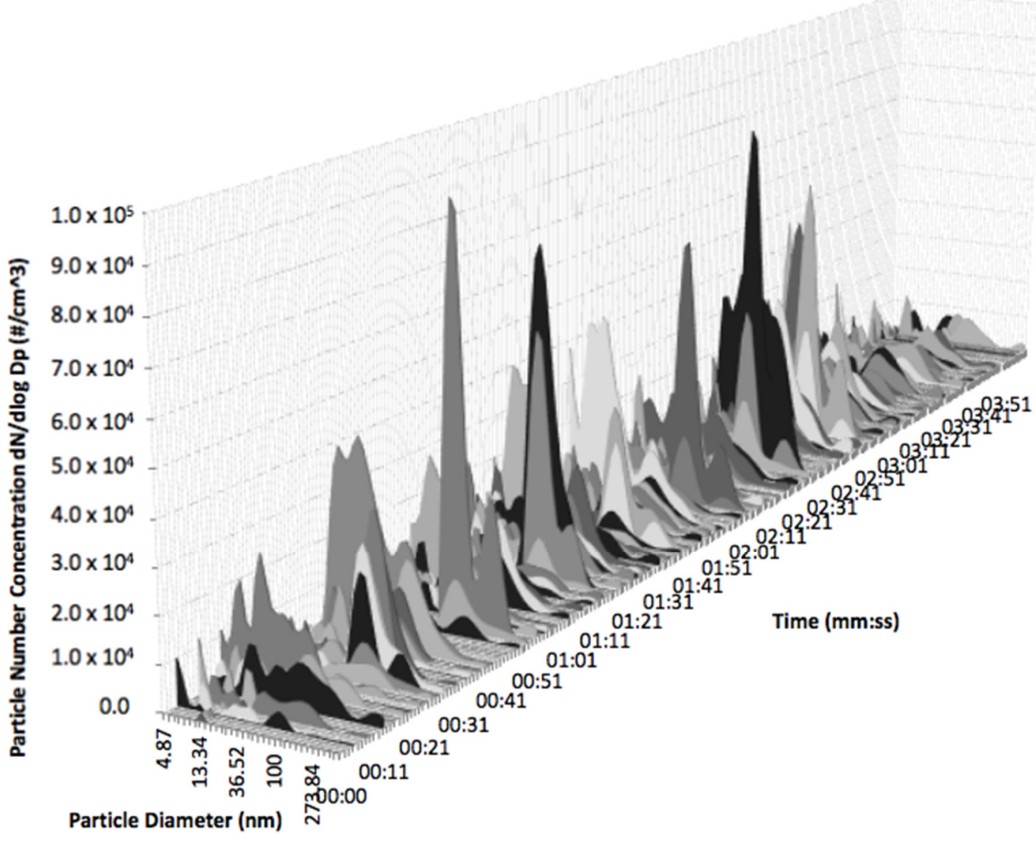

**Figure 6.** Particle size distribution recorded on the DMS50 during 4 min from the EP/CF/GO 0.1 wt.% sample.

As demonstrated in the CPC data previously shown in Figure 3, the DMS50 data display the peaks introduced during drilling across the first three minutes, followed by the post-drilling minute with reduced particle number concentrations. The eight peaks represent the eight holes drilled. This is apparent in all of the samples reinforced with GO, although also slightly less obvious. Due to the relatively low particle number concentrations, the peaks during drilling are less apparent. Similarly, the particle size distribution is not clearly evident in the one-minute post-drilling and the concentrations between drilling. The difference between the samples is similar to the CPC data, where the EP/CF/GO 0.5 wt.% sample exhibited the highest particle number concentration, which is conveyed into the DMS50 data. Lower peak concentrations are observed for the other samples, with relatively lower concentrations after drilling. As a result, the DMS50 data concur with the CPC data on the influence of the GO on particle number concentration.

It is noticeable within all samples, and as demonstrated in the CPC data, that the peak particle number concentrations introduced during the drilling are relatively inconsistent, followed by a more stable and consistent post-drilling concentration. Although the particle size distributions introduced during the peaks from drilling are different between samples, the distributions are relatively consistent within each sample. The particle size distributions can therefore be associated to the material, as opposed to any factor related to the continuation of the drilling, such as the particles present or the number of holes already been drilled by the drill bit. A two-dimensional plot of the average particle size distribution introduced at the peaks due to drilling will therefore be representative of the eight holes drilled for each sample and is presented in Figure 7.

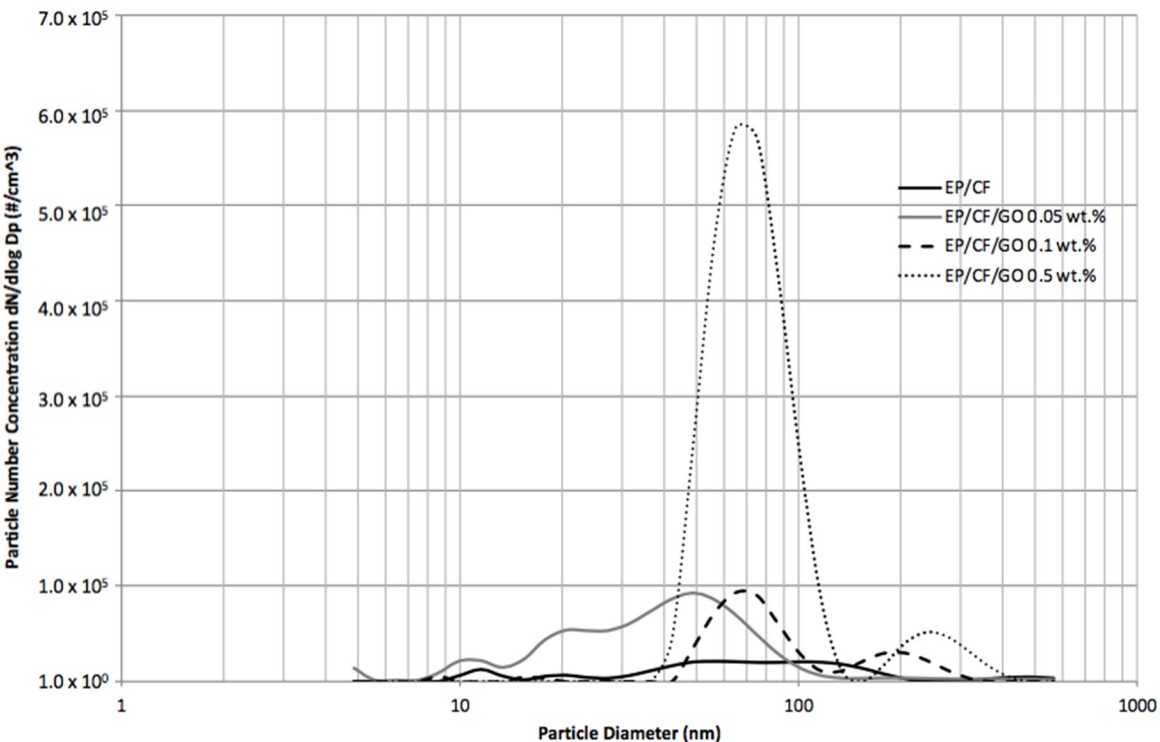

**Figure 7.** Peak particle size distribution within the 4 min sampling of the EP/CF-based samples reinforced with GO recorded on DMS50.

The four samples displayed contrasting results in particle size distribution released at the moment of drilling. Although demonstrating different peak sizes, a relatively high proportion of the size distribution from the four samples is ascertained to be between 40 and 100 nm in particle diameter. Whilst the EP/CF sample did not display a discrete sharp peak, all other samples revealed the highest peak within this same size range. This is, however, the one similar element visible in the four

samples. The sample without any GO nanoparticles observed a broad range of particle diameters. Similarly, the peaks exhibited from the EP/CF/GO 0.05 wt.% were split across a 10 to 70 nm particle diameter. In contrast, the EP/CF/GO 0.5 wt.% displayed two peaks, one between 40 and 100 nm and one between 150 and 400 nm in particle diameter.

In comparison to the SMPS data of the particle size distribution, the broad range and variation in particle diameter for the EP/CF, EP/CF/GO 0.05 wt.% and EP/CF/GO 0.1 wt.% samples are in moderate agreement with the DMS50 data. The peak observed at around 100 nm from EP/CF/GO 0.5 wt.% sample is somewhat similar to the SMPS data, however, the DMS50 did not display a peak for the sample at lower concentrations. Similar to the data presented in previous chapters, the disparate peaks seen on the two instruments introduce debatable deductions and limited effectiveness of instrumentations required for real-time data. Nonetheless, the data from the DMS50 demonstrated no evidence of independent nano-sized GO fillers within any of the particle size distributions for the GO reinforced samples. With almost no peaks apparent with a particle diameter less than 10 nm, the suggestion of GO layers separation due to the drilling shear forces is not evident in the DMS50 data. However, the GO is shown to increase the particle number concentration between 40 and 100 nm. The source of the increase is the higher particle number concentration observed in the GO reinforced samples. However, due to the particle size diameters, these cannot be associated to independent nanofillers, and instead either agglomerations or matrix-filler embedded particles. Nonetheless, all three instruments used to quantify the released particles (CPC, SMPS and DMS50) demonstrate a harmonized maximum increase in particle number concentration from the EP/CF/GO 0.5 wt.% sample.

*3.4. The Filler Effect on Mass Size Distribution*

Since the drilling was conducted without any interference from background particles, all of the particles measured on the instrumentation are from the nanocomposite material. With the use of the SMPS and the known density of the individual nanocomposites, the particle mass concentration can therefore be estimated. The data utilize the diameter of the particles measured using the SMPS. The constant material density for the three nanocomposites are: EP/CF = 1.59 g/cm$^3$, EP/CF/GO 0.05 wt.% = 1.59 g/cm$^3$, EP/CF/GO 0.05 wt.% = 1.59 g/cm$^3$ and EP/CF/GO 0.5 wt.% = 1.57 g/cm$^3$. The average mass concentration across the 4-min sampling period for different particle size diameters is illustrated in Figure 8.

The particle diameters with high particle number concentrations observed in the SMPS results have been adjusted due to the consequent mass increase of larger particles. Almost no significant peak was perceived below 50 nm. All of the samples consequently displayed an increase in particle mass concentration in diameters larger than 50 nm up until the SMPS limit of approximately 157 nm. As with the particle size distribution, the EP/CF/GO 0.5 wt.% demonstrated the largest peak at around 100 nm. The remaining samples recorded similar relative peaks between 50 and 157 nm. The EP/CF/GO 0.05 wt.% and EP/CF sample displayed a similar increasing profile in particle mass concentration over 100 nm. As with the particle number concentration and particle size distribution, the EP/CF/GO 0.5 wt.% clearly demonstrated an augmenting effect in concentration, with similar mass concentrations for the remaining EP/CF, EP/CF/GO 0.05 wt.% and EP/CF/GO 0.1 wt.% samples.

Since the CPC can measure a larger particle size range, an alternative mass concentration is valuable to quantify the release. Using the particle number concentration measurement at the end of the four-minute sampling period, and the calculated total quantity of mass drilled, an estimation of the concentration of particles/mass drilled can be acquired and is presented in Figure 9. This is calculated using the particle number concentration of the CPC (size range: 7 to 3000 nm), material density values and equivalent of mass drilled based on hole size and number of holes.

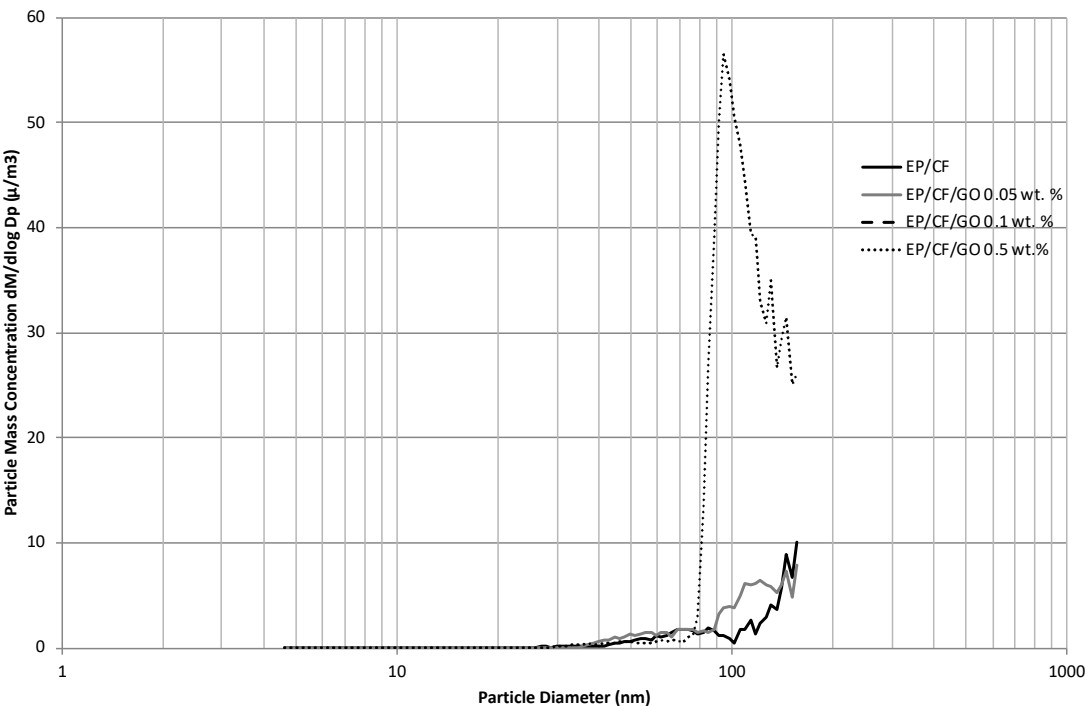

**Figure 8.** Particle mass concentration average over 4 min of EP/CF based nanocomposites reinforced with GO determined from SMPS (*n* = 12 for each average).

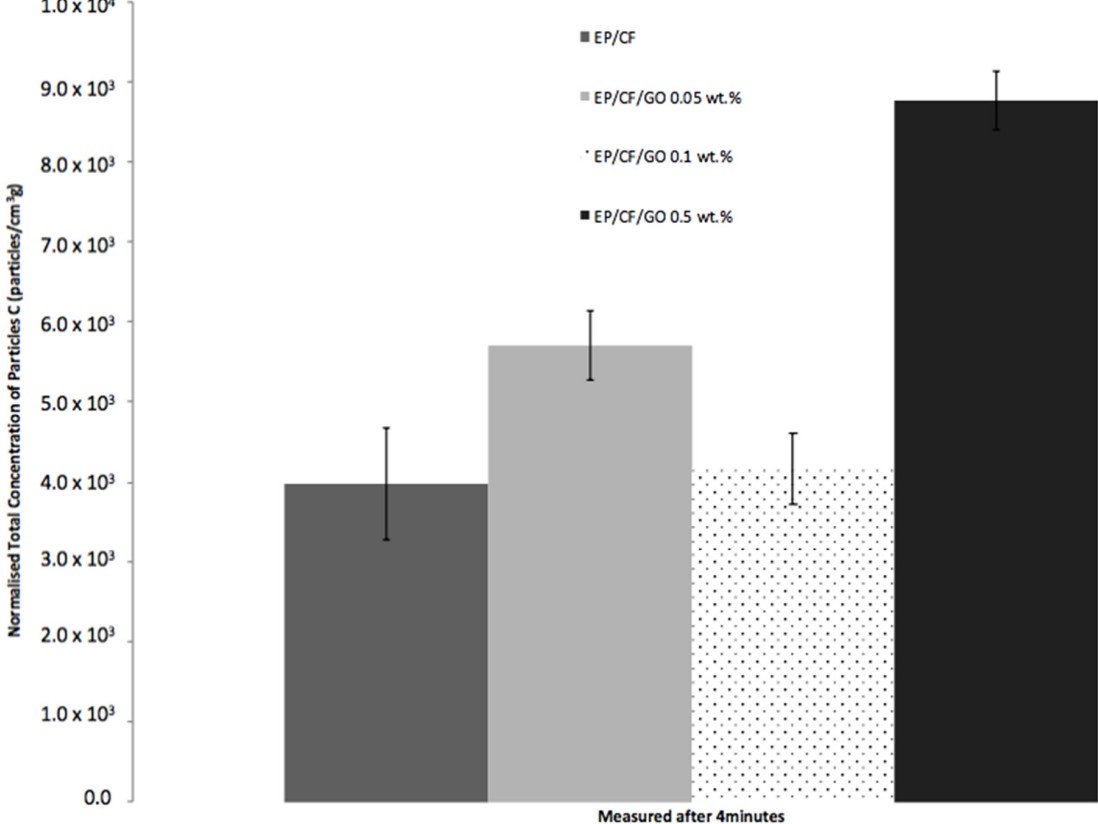

**Figure 9.** Normalized total concentration of particles (C divided by estimated drilled mass) recorded at 4th min for EP/CF based samples reinforced with GO (*n* = 3 for each average).

The number of particles to mass drilled ratio also presents the EP/CF/GO 0.5 wt.% sample with the highest particle release over the EP/CF and other GO reinforced samples (EP/CF = 3974 $\#/cm^3 g_{drilled}$, EP/CF/GO 0.05 wt.% = 5702 $\#/cm^3 g_{drilled}$, EP/CF/GO 0.1 wt.% = 4167 $\#/cm^3 g_{drilled}$, and EP/CF/GO 0.5 wt.% = 8758 $\#/cm^3 g_{drilled}$). Since the density of the EP/CF/GO 0.05 wt.% and EP/CF/GO 0.1 wt.% sample did not change sufficiently to be measured with the addition of the GO, the correlation to the EP/CF sample is the same as the particle number concentration previously presented. However, the slight decrease in density in the EP/CF/GO 0.5 wt.% sample means that the sample observed a 118% increase in normalized total concentration over the EP/CF sample.

The particle mass concentration is an important identified parameter within the literature when evaluating the release or exposure to nanoparticles. The data identify important differences and support the findings on the effect of the filler on particle number concentration and particle size distributions. The GO at lower weight concentrations shows minimal effect on the release, whereas the EP/CF/GO 0.5 wt.% sample displayed a significant difference in comparison to the EP/CF sample.

### 3.5. Assessment of Deposited Particles

The debris collected in the chamber as described in the methodology was analyzed using an SEM. An SEM image of the debris for each sample is displayed in Figure 10. A larger magnification of the dust collected in the sampling placed underneath the drilling is shown in Figure 11.

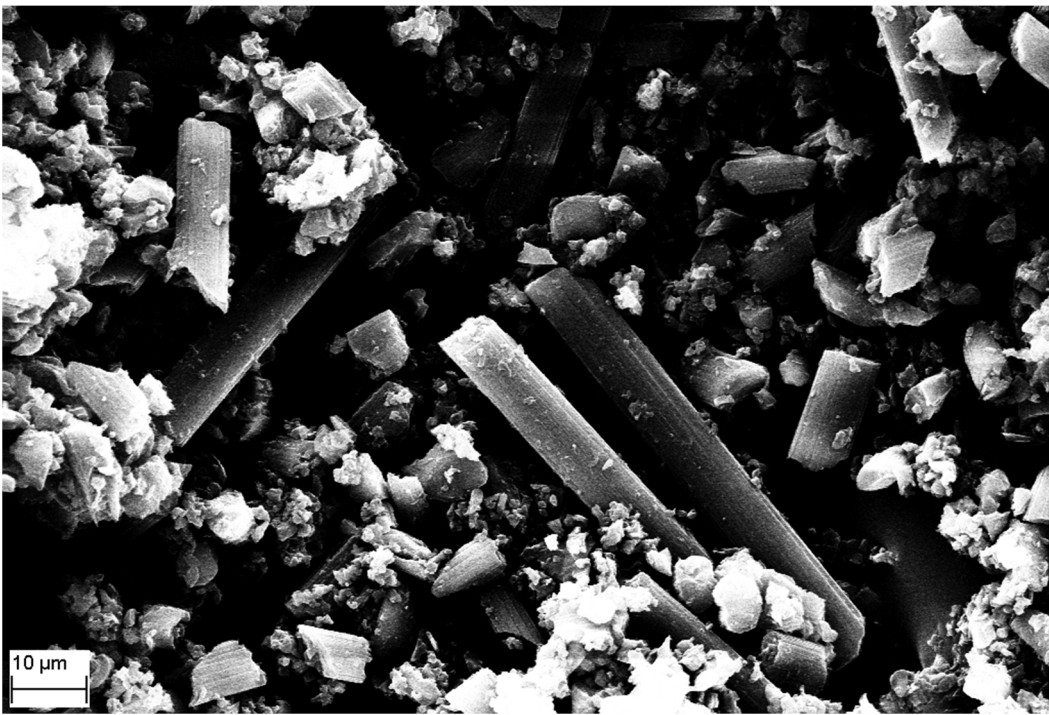

**Figure 10.** Deposited particles collected in a sampling tray placed directly below drilling from the EP/CF/GO 0.5 wt.% sample.

The deposited particles collected illustrate a large variation, such as particles, agglomerates and independent fibers and matrix. The image has a relatively distant magnification, which allows us to display the micro-sized CFs and particle aggregation at a micro level. The nanoparticles are therefore not distinguishable and are shown in Figure 11.

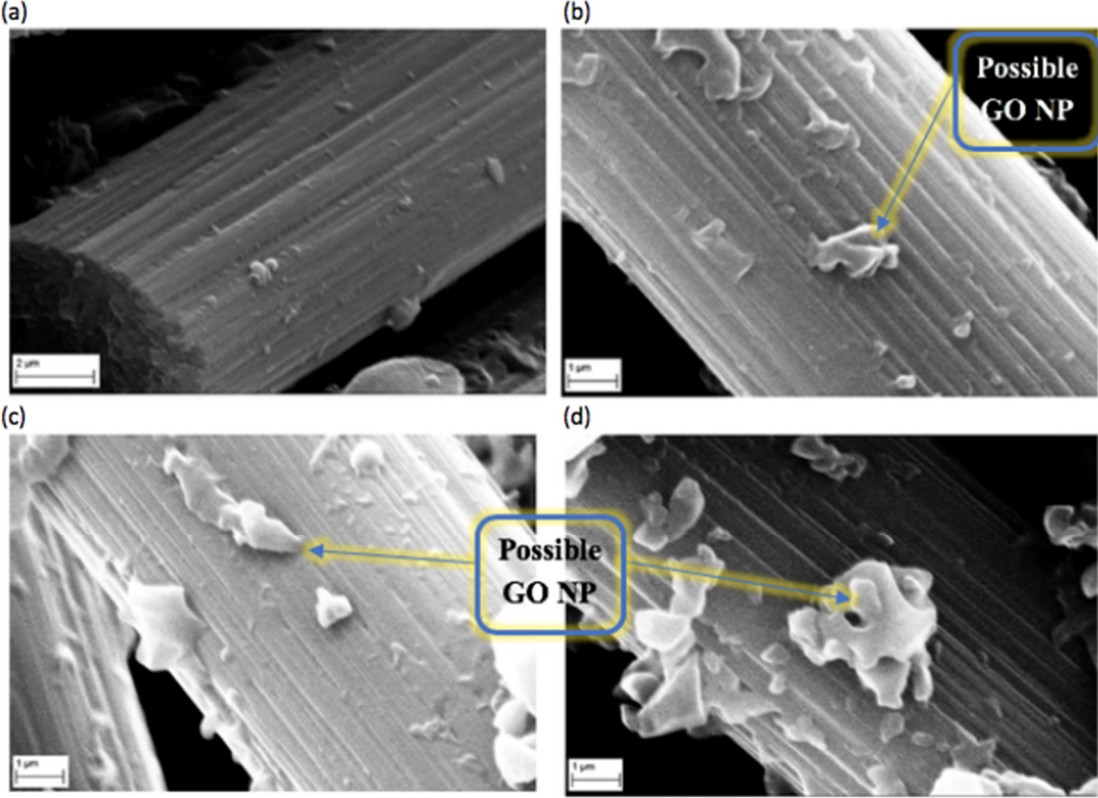

**Figure 11.** SEM images of deposited particles collected in sampling tray from drilling on (**a**) EP/CF sample, (**b**) EP/CF/GO 0.05 wt.% sample, (**c**) EP/CF/GO 0.1 wt.% sample and (**d**) EP/CF/GO 0.5 wt.% sample.

Within the microscopy analysis of all the GO reinforced samples, no independent GO nanoparticles were identified. The GO reinforced samples were instead seen to demonstrate an increase in particles embedded or adhered to the surface of the CFs, as can be seen in Figure 11. The neat EP/CF sample displayed significantly fewer particles attached onto the independent fibers identified within the deposited particles. The few particles attached onto the CF shown in Figure 11a can be attributed to the EP, as no GO has been added. The GO-reinforced samples, however, demonstrated visibly more particles on the surface of the CFs. This can be attributed to either the GO particles or the EP. As is observed in the literature, GO particles are seen to improve the interfacial bonding between the CF and the EP. The microscopy images of the surface of the carbon fibers with attached particles of GO and EP are in accordance with similar findings in other recent studies that embedded GO within EP/CF samples [49,50]. The increase in the number of particles attached to the deposited fibers was the only identifiable difference in the morphology studies investigating the influence of the GO filler weight concentration. The deposited particles therefore observed no identifiable independent GO nanoparticles, and instead were seen to increase the particles bonding to the surface of the CFs.

The findings within deposited particles, therefore, do not aid in identifying the source of the noted increase in the particle size distribution and particle mass distributions at 100 nm. Further, the deposited particles do not provide evidence of independent GO nanoparticles released from the embedding within the nanocomposite materials. The data, however, are a representation of the deposited particles collected within the sampling tray, and not the measured airborne particles through the particle quantification instruments. Within the deposited particles, the release is shown to be matrix or CF-orientated with GO embedded or adhered to the surface.

## 4. Discussion

### 4.1. Influence of Filler

From the three samples investigated with the incorporation of nanofillers, all three demonstrated a statistically significant difference in a two-sample t-test of significance of each sample mean and deviation to the reference sample (test of 90% confidence interval). It is important to note that although the statistical analysis returned a statistically significant difference with the introduction of the GO, this does not include the extent of the difference. The incorporation indicates a minor influence in the increase in particle number concentration. With the comparison of the samples, the EP/CF/GO 0.5 wt.% demonstrated a clear increase in all aspects of the particle number concentration; whereas the 0.05 and 0.1 wt.% displayed a more minor increase in peak particle number concentration values. As with all other samples, the statistical analysis does consider the high standard deviation and range demonstrated and therefore includes the level of randomness and variability in the peaks released.

However, the different weight concentrations of GO within EP/CF samples revealed that there is no direct correlation between weight concentration within the nanocomposite and influence on particle number concentration. The samples displayed an increase in particle number concentration with 0.5 and 0.05 wt.% GO, but a minimal effect with 0.1 wt.% GO (EP/CF/GO 0.05 wt.% = 43% increase, EP/CF/GO 0.1 wt.% = 5% increase, and EP/CF/GO 0.5 wt.% = 118% increase).

In view of the fact that the GO reinforced samples demonstrating the filler weight concentration within the nanocomposite alone does not correlate to the influence particle number concentration (i.e., increase in weight concentration does not demonstrate an equivalent increase in particle number concentration), the influence on particle number concentration can be compared with the influence on mechanical properties. With all other parameters unchanged, the only change in parameter is the nanocomposite composition. The comparison between the influence of reinforcement fillers with the EP/CF without GO reinforcement is shown in Figure 12.

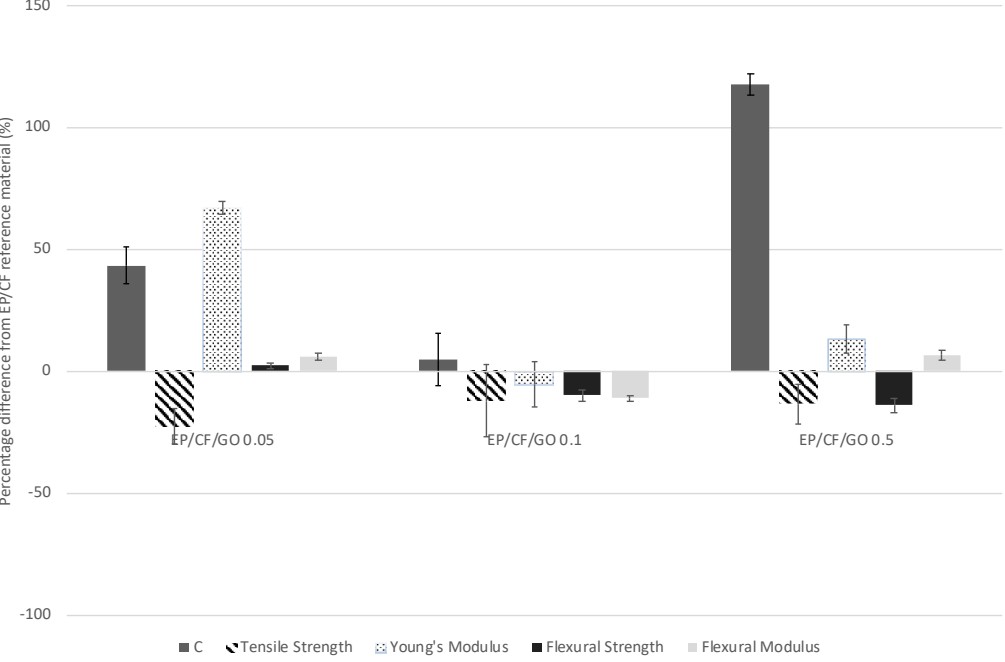

**Figure 12.** Comparison of percentage difference to neat EP/CF and therefore, the influence of GO within EP/CF-based samples on particle number concentration (C), tensile strength, Young's modulus, flexural strength and flexural modulus (Note high standard deviations are observed due to the combined deviations of each sample and EP/CF).

The use of GO as a nanofiller displayed conflicting effects on the mechanical properties and particle number concentration from the EP/CF hybrid composites. The EP/CF/GO 0.05 wt.% sample displayed the most significant improvement in mechanical properties with a statistically significant increase in flexural strength, flexural modulus and Young's modulus in comparison to the EP/CF sample. This resulted in a statistically significant increase in particle number concentration, but lower than the increase introduced from the EP/CF/GO 0.5 wt.% sample. The 0.1 wt.% GO observed a decrease in all mechanical properties, whilst the 0.5 wt.% GO displayed a statistically significant increase in both Young's and flexural moduli but a decrease in related strengths. These results contrast with several studies that showed an increase in mechanical properties with low weight concentrations of GO and a decrease after a threshold quantity. Another study [51] found the peak mechanical performance at 0.3 wt.% GO with a clear decrement in properties from 0.4 wt.% GO. In contrast, a different study [15] observed maximum flexural properties at 0.5 wt.% GO, followed by a decline. However, no studies were found to have reported on the mechanical properties with concentrations lower than 0.1 wt.% GO added to EP/CF and this is therefore the first reporting of such. Similarly, no studies have reported on the release of GO from EP/CF hybrid composites and this is therefore also the first reporting of such to the authors best knowledge.

Among the literature, the improvement to the bonding between the CF and polymers has shown that oxidative treatments or particles that can generate –OH or –COOH groups on the fiber surface will act as coupling or bonding agents [52]. GO can effectively enhance the interfacial adhesion as the sp2 structure of the GO is prone to attach onto the surface of the CF by π–π stacking interaction [53]. The improved bonding and interfacial adhesion between the matrix, GO and CF allows for an optimization in stress transfer between the softer matrix of the polymer phase, to the CF [2]. The limit of GO content is said to be at the point where GO initiates to bond with the hardener and hence prevent the interface between the epoxy and hardener. The cross-linking therefore is reduced, resulting in a weaker interfacial interaction [51]. The peak concentration was not evident in either the material properties or nanoparticle release. The 0.1 wt.% GO sample demonstrated the least influence on mechanical properties and also nanoparticle release. However, improvement in mechanical properties observed from the 0.05 and 0.5 wt.% GO samples did not show a correlation to the particle number concentration increase. Therefore, although the GO nanoparticles can be seen to influence the particle number concentration and mechanical properties, no correlation between weight concentration and the subsequent mechanical properties or nanoparticle release are evident.

As discussed, currently only nine studies have evaluated the influence of nanoparticles on nanoparticle release during drilling [23–31]. All of the studies observed nanoparticle release and highlighted the potential hazard and exposure to humans which needs to be understood. As with the conclusions within the literature on the influence of nanoparticles on nanocomposite material properties [54], the correlation between nanoparticles and effect on nanoparticle release is challenging. The results are in unison with multiple reports in the literature, as eluded to earlier on, in the complexity of nanoparticle release data from the nanoparticle fillers. A noteworthy observation by Hankin and Read [55], appropriate to the findings within this study, however relating to the current knowledge of risks associated with nanotechnology, noted that "research conducted to date has shown the potential risks of nanotechnologies to be associated with a high degree of complexity and uncertainty, with no clear-cut cause-and-effect relationship".

Although there is currently no predictive model of the release of nanoparticles from nanocomposite during drilling, the closest literature is on the production of emissions from drilling in metals [56–58]. The studies reported the fracture of the material to be highly associated to the brittleness of the material. The GO nanoparticles used within this study had minimal influence on the brittleness and ductility of the materials in comparison to the reference EP/CF without the nanoparticle fillers. Although the nanoparticles were observed to have an influence on some of the material properties, overall no significant influence on the brittleness of the material was observed, which might explain the lack of clear correlation observed in nanoparticle release with increasing weight concentration of GO.

### 4.2. Influence of Matrix

As the same methodology was used in three other studies [28–30], the materials used across the studies can be compared. In contrast to EP-based samples from one study [29] without CF reinforcement, the peak particle number concentrations in this study can be seen to be significantly lower. Furthermore, a comparison with the materials utilized in other studies [23–27,31] highlights a significant contrast in particle number concentrations observed across the studies. Ranging from a peak concentration of around $7.5 \times 10^3$ #/cm$^3$ [28] compared to peaks of over $2.2 \times 10^9$ #/cm$^3$ [27], this demonstrates how the polymer might be a major factor in the particle number concentration, with differences of up to $10^6$ #/cm$^3$. However, this should also be taken with caution, as studies have used different drilling setups, parameters, lab environments and measurement techniques.

Nonetheless, evaluating the general material brittleness and ductility (i.e., correlation to point of failure and plasticity observed) from the mechanical properties associated to the groups of materials used, a closer correlation to the nanoparticle release can be acknowledged. The interaction between the drill and material at the microstructure level effect the form of the chips created. Therefore, the material fracture mechanics and plasticity deformation could help to predict how the material will behave at the local stress levels induced by the drilling and can thus be associated with the characteristics of nanoparticle release. In comparison to another study [28], a thermoplastic and more ductile properties (polypropylene) observed a much lower particle number concentration than the brittle thermosets of EP/CF samples observed within this study. The high energy release when subject to stress of the brittle materials causes a considerably higher particle number concentration. The material ductility therefore indicates an influence on the nanoparticle release. This is similar to the conclusions in the studies on metallic material drilling [56–58].

Although the introduction of the nanoparticles at the varied weight concentrations within this study is shown to have an effect on the particle release in comparison to the EP/CF, the basic profile of the release did not observe a significant change (unlike a comparison of the data with other studies). Therefore, whilst the nanoparticles might have an effect on the nanoparticle release, the substantial profile of the release is indicated to be dictated by the polymer and larger filler weight concentrations. This could be compared to the influence the nanoparticles have on the mechanical properties. Whereas the nanofillers were shown to have an influence on the mechanical properties (i.e., up to 65%), most of the mechanical properties are inherited from the EP/CF without the nanoparticle reinforcement. Due to the low weight concentrations, the mechanical properties are highly driven by the matrix, which can be seen in the nanoparticle release.

If the release is matrix oriented, this might have an effect on the health concerns arising from the use of the nanoparticles. Although some studies have demonstrated no increased toxicity [59–61], there is still a lack of understanding whether most embedded nanoparticles within the matrix are toxic, as they have not been investigated due to the complexity and variations in material phases [19,20]. The toxicity studies previously mentioned within this study report the understanding and toxicity of only the individual GO nanoparticles as opposed to a matrix/filler combination. Additionally, the identification of release of the embedded hazardous nanoparticles must also be linked to the exposure of the released particles for toxicological assessments [38]. As with particles exposed to human cells through inhalation, and ingestion, the literature has reported it to be necessary to study each nanoparticle individually to fully understand the toxicity effects [62,63]. The particle size has been highlighted to be one of the most influential material properties affecting toxicity, but is one of only many physio-chemical properties, such as shape, aspect ratio, surface area, chemical composition and crystalline structure, of nanoparticles that have been identified to influence the nanoparticle toxicity [63]. Additional to the concentration or potential dosage, this study reports the particle size, mass and morphology, which can all contribute to the toxicity [43].

As the matrix can be associated to have a significant effect on the particle number concentration, the particle size distributions observed similar dependency on the matrix. The use of the GO fillers introduced minor shifts in peaks with the introduction of the diameters.

Furthermore, the data presented within this study are a representation of particle release from a process-related approach [38]. The methodology used a clean environment through the removal of all background interference. The data collected are therefore a representation of the particle emissions solely from the material. Removing the background data allows for a depiction of any particles released from the materials which can be directly linked as an unconditional maximum exposure assessment [38]. The results represent a worst-case scenario of potential nanoparticle atmospheric emissions from the materials. The removal of any background particles provides a clean environment to be able to evaluate the full release from the investigated materials. Particle background interference will differ in all lab environments and could influence/affect the particles release. The data provided allow for a comparison and evaluation of the material with and without the nanoparticle fillers and can be used to identify whether release is likely. The full extent of exposure or intensity in a workplace scenario or influence with atmospheric air could potentially differ and should therefore be evaluated. The results therefore represent the potential release of the fillers and do not represent the exposure. As discussed, in the literature [18], the identification of potential release is necessary in relation to the materials and given scenario. Other exposure determinants that may be important, such as personal behavior, experience, maintenance of hoods/ventilations, as well as housekeeping practices, will need to be considered when using the data to determine any exposure controls.

## 5. Conclusions

The aim of this study is to investigate the influence GO has on nanoparticle emissions from EP/CF hybrid ENMs during drilling. Four EP/CF based composites were manufactured with three variations in weight concentrations of GO; 0.05, 0.1 and 0.5 wt.%. The influence of the GO nanoparticle weight concentrations has on nanoparticle release during drilling was investigated and correlated to the influence on mechanical properties. All samples demonstrated nanoparticle release, including the neat EP/CF sample without any GO nanoparticles. The results from the study offer a novel set of data comparing the nanoparticle release of GO with varying nanoparticle filler weight concentration and correlating it with the mechanical influence of the fillers. From the results, different conclusions were deduced.

All three nanoparticle reinforced samples demonstrated a statistically significant difference in comparison to the EP/CF sample. The GO-reinforced samples demonstrated that the filler weight concentration within the nanocomposite alone does not correlate to the influence particle number concentration (i.e., increase in weight concentration does not demonstrate an equivalent increase in particle number concentration). Although a two-sample t-test of significance of each sample mean and deviation to the neat EP/CF sample returned statistically significant differences for all concentrations of GO (outside the 90% confidence interval), the inclusion of 0.05 and 0.1 wt.% GO nanoparticles demonstrated minimal effect on nanoparticle release. However, the EP/CF/GO 0.5 wt.% demonstrated a 243% increase in mean peak particle number concentration introduced during drilling. Similarly, at the end of the four-minute sampling period, the EP/CF/GO 0.5 wt.% sample observed a 118% increase in comparison to the EP/CF sample. However, the minor increases observed for the lower weight concentrations of GO reinforced samples do not substantiate an increase in particle number concentration with an increase in GO nanoparticles (EP/CF/GO 0.05 wt.% increase of 44% and EP/CF/GO 0.1 wt.% increase of 5% from the EP/CF sample measured at the 4th minute). Nonetheless, the statistical analysis returned a statistically significant difference with the introduction of GO nanoparticles within the nanocomposites on nanoparticle release during drilling. The particle size distribution illustrated minimal effect with the introduction of GO nanofillers in comparison to the EP/CF sample. Due to the particle size diameters, the peaks cannot be associated to independent GO nanofillers, and instead either agglomerations or matrix-filler embedded particles. Correspondingly, the assessment of the deposited particles displayed no evidence of independent GO nanoparticles.

The comparison between the nanoparticle release and mechanical properties demonstrated no observable correlation with the samples used within this study. However, the comparison of the

results with other studies highlights the potential for a correlation of the nanoparticle release and the matrix material properties. More precisely, the comparison between matrices highlighted that the majority of the release characteristics were indicated to be dependent on the material polymer brittleness. A comparison of the data leads towards a conclusion that the more brittle the material is, the higher the particle number concentration. The relatively minor influence that the GO nanoparticles have on mechanical properties exhibit similar minor, yet still statistically significant, influences on the nanoparticle release.

Nonetheless, although the GO nanofillers are concluded to have demonstrated an influence on the nanoparticle release during drilling, the materials demonstrated a level of complexity with no clear cause and effect relationship. However, the possible dependence on polymer properties and correlation between the nanoparticle release and matrix mechanical properties may be used when improving materials and has the potential to act as part of developing nanocomposite materials as a concept of safety by design.

**Author Contributions:** Conceptualization, K.S. and J.N.; methodology, K.S. and J.N.; software, K.S. and J.N.; validation, K.S. and J.N.; formal analysis, K.S. and J.N.; investigation, K.S. and J.N.; resources, K.S. and J.N.; data curation, K.S. and J.N.; writing—original draft preparation, K.S. and J.N.; writing—review and editing, K.S. and J.N.; visualization, K.S. and J.N.; supervision, J.N.; project administration, J.N.; funding acquisition, J.N. All authors have read and agreed to the published version of the manuscript.

**Funding:** The work is funded by and part of the European Commission Life project named Simulation of the release of nanomaterials from consumer products for environmental exposure assessment (SIRENA, Pr. No. LIFE 11 ENV/ES/596). We are also thankful to the funding by QualityNano Project through Transnational Access (TA Application VITO-TAF-382 and VITO-TAF-500) under the European Commission, Grant Agreement No: INFRA-2010-262163 for the access and use of the facilities at the Flemish Institute for Technological Research (VITO).

**Acknowledgments:** The authors are grateful to the funding bodies and would also like to acknowledge E Frijns, J Van Laer, K. Tirez and R. Persoons at VITO NV (Mol, Belgium) for their support during nanoparticles release measurements, XRF and SEM characterisation.

**Conflicts of Interest:** The authors declare no conflict of interest. The funders had no role in the design of the study; in the collection, analyses, or interpretation of data; in the writing of the manuscript, or in the decision to publish the results.

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
