# Peer review of "The Influence of Graphene Oxide on Nanoparticle Emissions during Drilling of Graphene/Epoxy Carbon-Fiber Reinforced Engineered Nanomaterials"

_atmosphere, doi:10.3390/atmos11060573_

Round 1
Reviewer 1 Report
Dear Authors.
I would like to stress that the study You describe seems to be important, as not all of the issues regarding wear or process induced nanoparticle emissions are widely known or considered on the design stage of a product lifecycle.
As I understand You assume that the naoparticles (around 20 nm) are GO particles emitted during testing. Is it only the assumption? Or there is some prove that these are GO NPs?
I do not like the lines between the experimental points in the Figure 2. There is no evidence that tensile and flexural properties vary in that way. Also it seems to me that Go addition does not affect the mechanical properties of the composite. Is it because the composite was made just for testing purposes and no improvement in the properties was expected?
Is figure 6 necessary? It is not clear at all i n my opinion.
Figure 10 in my copy of the manuscript is of unacceptable quality.
In the figure 11 please make an indicative arrow to the NPs You associate with GO filler.
I would also lie to encourage the Authors to comment on the morphology of the NPs associated to GO filler. Probably it is not only size of the NP that defines its possible toxicity. I would like to see if the filler concentration results in different particles being emitted.
Most of all I would like to be sure that mentioning Graphene Oxide in the title is not just for making the manuscript sound more interesting.
Author Response
Reviewer 1:
Dear Authors.
I would like to stress that the study You describe seems to be important, as not all of the issues regarding wear or process induced nanoparticle emissions are widely known or considered on the design stage of a product lifecycle.
Thank you for this comment.
As I understand You assume that the nanoparticles (around 20 nm) are GO particles emitted during testing. Is it only the assumption? Or there is some prove that these are GO NPs?
Thanks for highlighting this. Yes, each sheet of the GO used can have a thickness of around 1 nm (796034 Sigma Aldrich) and since the GO embedded consists of 15 to 20 sheet flakes, the GO particles will therefore have a thickness of up to 20 nm. No increase in peaks around 20nm were observed, and the peaks seen below 6 nm (Figure 5) are the only evidence of possible layers of the GO fillers separated from the material. However, the particle size distribution cannot confirm this alone without either identification of the GO fillers in the microscopy or chemical analysis. The microscopy analysis of the deposited particles did not provide evidence of independent GO nanoparticles released from the embedding within the nanocomposite materials. Therefore, whilst the particle size distribution is showing possible separation of the GO NPs, there is no evidence of the separation in the microscopy studies of the deposited particles. Taken this into consideration, we have added to line 338-339 (highlighted in blue) and the above mentioned in lines 336- 341, 477-479 and 656-660.
I do not like the lines between the experimental points in the Figure 2. There is no evidence that tensile and flexural properties vary in that way. Also it seem1s to me that Go addition does not affect the mechanical properties of the composite. Is it because the composite was made just for testing purposes and no improvement in the properties was expected?
Thank you for this. The lines from the graph shown in Figure 2 have been removed, and Figure 2 replaced. And yes, the composites were made for testing purposes and to see any correlation between mechanical properties that are related to the fracture properties that might correlate to the emissions during drilling. As is stated in line 202-204, none of the samples returned a statistically significant difference in tensile strength, but only the EP/CF/GO 0.1 wt.% did not return a statistically significant difference for Young’s modulus. From the flexural properties, all of the samples returned a statistically significant difference in flexural modulus and only one (EP/CF/GO 0.05 wt.%) returned with a statistically insignificant difference in flexural strength. This is mentioned in lines 196 – 215.
Is figure 6 necessary? It is not clear at all i n my opinion.
Thank you for this comment. Although Figure 6 can be difficult to interpret, in demonstrates the change in particle size distribution every second across the experiment. The Figure helps support the evidence from the CPC data that high concentrations are introduced during the 8 holes drilled and the influence of GO, as well as an alternative instrument to see if NP in the size range of the GO are released. Together with the SMPS, the two equipment show no clear evidence of independent nano-sized GO fillers released.
Figure 10 in my copy of the manuscript is of unacceptable quality.
Thank you for this comment. Figure 10 has been replaced with a higher quality SEM image of the deposited particles collected in sampling tray placed directly below drilling from EP/CF/GO 0.5 wt. % sample.
In the figure 11 please make an indicative arrow to the NPs You associate with GO filler.
Thank you for this suggestion. Figure 11 has been replaced with indicative arrows to NPs possibly associated with the GO filler.
I would also lie to encourage the Authors to comment on the morphology of the NPs associated to GO filler. Probably it is not only size of the NP that defines its possible toxicity. I would like to see if the filler concentration results in different particles being emitted.
Thank you for highlighting this point. The authors agree with the reviewer, that the size alone does not define the possible toxicity. An extension in relation to this point has been added in lines 609 – 614 (highlighted in blue).
The morphology studies demonstrated an increase in particles embedded or adhered to the surface of the CFs with increase in the GO filler weight concentration. As would be expected, with a higher weight concentration of the GO, more particles were observed on surface of the fibers. This can be attributed to either the GO particles and/ or EP, as GO particles are reported to improve the interfacial bonding between the CF and EP. The filler concentration resulted in no further clear influence on the morphology studies on the deposited particles. A further comment has been added, line 477 – 479 in relation to this point.
Most of all I would like to be sure that mentioning Graphene Oxide in the title is not just for making the manuscript sound more interesting.
Thank you for this comment. A main focus of this study is to evaluate the influence of the different GO weight concentrations on the nanoparticle emissions. As GO is increasingly being used as a filler within composite materials, it is vital that any risk associated with the filler is also understood. Process induced nanoparticle emissions as a whole are not yet fully understood but are even more important when GO has shown conceivable health risks and toxicity to humans and the environment if released.
Reviewer 2 Report
The manuscript titled “Investigation of the influence of graphene oxide on nanoparticle atmospheric emissions during drilling from graphene/carbon-fiber reinforced epoxy engineered nanomaterials” is an interesting study about the emission of nanoparticles in EP/CF composites with and without GO during a common industrial process as is drilling. All experimental methodology is well described, and the discussion of the results is clear and really well compared with the literature. In my opinion, this manuscript can be published in Atmosphere. I have only a minor comment, in this EP/CF composite one of the most important factor for mechanical properties, if not the most, is the composition between EP/CF, I would like to ask if authors checked that the composition is constant after the addition of GO. The addition of GO affected surely the viscosity of the resin and therefore the infusion process, which may cause a variation in the concentration ratio between CF and EP. Really small variation in the composition has an important influence in mechanical properties. This could be an explanation for the increase of the Young Modulus with 0,05 wt% of GO compared to EP/GF, but with 0,1 and 0,5 wt% this increase is almost negligible, or even decrease as in the case of 0,1 wt%.
Author Response
Reviewer 2:
“The manuscript titled “Investigation of the influence of graphene oxide on nanoparticle atmospheric emissions during drilling from graphene/carbon-fiber reinforced epoxy engineered nanomaterials” is an interesting study about the emission of nanoparticles in EP/CF composites with and without GO during a common industrial process as is drilling. All experimental methodology is well described, and the discussion of the results is clear and really well compared with the literature. In my opinion, this manuscript can be published in Atmosphere. I have only a minor comment, in this EP/CF composite one of the most important factor for mechanical properties, if not the most, is the composition between EP/CF, I would like to ask if authors checked that the composition is constant after the addition of GO. The addition of GO affected surely the viscosity of the resin and therefore the infusion process, which may cause a variation in the concentration ratio between CF and EP. Really small variation in the composition has an important influence in mechanical properties. This could be an explanation for the increase of the Young Modulus with 0,05 wt% of GO compared to EP/GF, but with 0,1 and 0,5 wt% this increase is almost negligible, or even decrease as in the case of 0,1 wt%.”
Thank you for this comment and suggestions. We agree with the reviewer that the EP/CF is the major factor to both the mechanical properties and nanoparticle emissions, along with the bigger influence and increase with the lower weight concentration of 0.05 wt.% GO. In relation to the variation in composition or concentration ratio between the CF and EP, any consequence would be represented in the mechanical or emission results. Having said this, we have also minimised any variation in the concentration between CF and EP, but agree that this might change with the addition of the GO into the EP. The concentrations of GO are relative to the wt.% of the quantity of EP prior to any manufacture or mixing with the hardener. The vacuum resin infusion process is a renowned method in maximising the fibre/resin volume fraction, and hence the justification for this method. Due to the vacuum created and the same quantity of resin infused, the process produces a 60/40 fibre/resin ratio without the consideration of GO. Since these values are such a low percentage (0.05 wt.% to 0.5 wt.%) of the resin content, and although this might affect the viscosity during mixing, this will have negligible effect on the fibre/resin ratio once the EP has cured. However, we recognise that we have not incorporated this and will evaluate the effect on viscosity in future studies. We added to the Experimental section to include this within the paper in lines 98 and 105-106.
Round 2
Reviewer 1 Report
Dear Authors,
I am glad some of my suggestions have been taken into account and the manuscript was accordingly revised.
I am still not 100% sure that NPs you are observing are GO NPs (SEM images do not show layered structure). However all the other results point towards you may be right.
Overall i recommend the manuscript for publication if the Editor decides so.